# SEMI-SUPERVISED POSE ESTIMATION WITH GEOMETRIC LATENT REPRESENTATIONS

## ABSTRACT

Pose estimation is the task of finding the orientation of an object within an image with respect to a fixed frame of reference. Current classification and regression approaches to the task require large quantities of labelled data for their purposes. The amount of labelled data for pose estimation is relatively limited. With this in mind, we propose the use of Conditional Variational Autoencoders (CVAEs) Kingma et al. (2014) with circular latent representations to estimate the corresponding 2D rotations of an object. The method is capable of training with datasets that have an arbitrary amount of labelled images providing relatively similar performance for cases in which 10-20% of the labels for images is missing.

## 1   INTRODUCTION

Pose estimation is the computer vision task in which the rotation of an object is estimated with respect to a certain frame of reference. The estimation of the position and the pose of an object within an image is an important challenge of computer vision and can be applied to a wide range of areas such as car navigation (Braun et al., 2016), augmented reality (Marchand et al., 2016) and robot interaction (Xiang et al., 2018).

Several methods based on the use of convolutional neural networks have been proposed high performance. These approaches are divided into two: classification-based, where the possible rotations that can be predicted are discretized into predefined sets with different coarseness levels and regression-based where a continuous prediction is provided.

Most of them are supervised methods which require large datasets with accurately labelled images and with a sufficiently good coverage of all possible rotations of each object instance. One of such datasets obtained from natural images and manually annotating objects in images is the Pascal3D+ dataset (Xiang et al., 2014) with about 22 thousand images from twelve categories. Compared to other datasets for other computer vision tasks such as classification (with millions of images) Li et al. (2009) the example data is scarce and the labelling process might be cumbersome and prone to errors. Thus, the need for a method to train with a limited amount of labelled data.

To solve this problem there have been dataset generation approaches proposed to create new training labels and images (Su et al., 2015; Varol et al., 2017; Johnson-Roberson et al., 2017) from 3D models and inserting them in natural scenery. Other approaches are focused on reducing the supervision to auxiliary tasks indirectly related to the pose estimation such as in Esteves et al. (2018b).Prokudin et al. (2018) addresses the problem of label quality by including an uncertainty estimation in the rotation predictions.

The work presented in this paper proposes a semi-supervised probabilistic method for pose estimation that incorporates the geometry of 2D rotations into the latent variable representations. The method is capable of using both labelled and unlabelled images from an object to improve the rotation regressions by performing the auxiliary task of reconstruction from the autoencoders. We have also investigated the effects of having a variable amount of labelled data in a controlled setting and variable coverage of the rotations.

In the general setting of object pose estimation an object is located in the scene of interest and an image has been taken of it. The object has a certain rotation with respect to the a fixed frame of reference. The goal is to estimate the rotation that the object has undergone from the available

image. In particular we are interested in the case were we might have a dataset in which the objects might be partially labelled.

More formally, consider that we have a training dataset consisting of $L$ labelled images with their corresponding rotation representations $\{(x_l, r_l)\}_{l=1}^{L}$ in which $x_l \in X$ and $r_l \in R$ correspond to the $l$-th image and rotation, respectively, and $U$ unlabelled images $\{x_u\}_{u=1}^{U}$. How can we obtain a model for estimating the pose of an object based on an image, taking advantage of both datasets? How does such a model perform with respect to the amount of supervision provided and the angle coverage?

## 2 RELATED WORK

Current approaches for rotation estimation using neural networks can be divided into classification and regression. Classification methods discretize the possible rotations orientations of an object (Su et al., 2015; Li et al., 2019b; Tulsiani & Malik, 2015) while regression methods provide a more approach to the continuous nature of rotations (Mahendran et al., 2018a;b; Liao et al., 2019; Prokudin et al., 2018).

Since rotations have a particular topology, their structure needs to be taken into account for both classification and regression tasks. Mahendran et al. (2018a) studies the performance of regression methods by using different rotation representations in a supervised setting. Liao et al. (2019) focus on the task of regression by representing rotations in $n$-spheres for different pose-related tasks. Prokudin et al. (2018).

In Esteves et al. (2018b;a) equivariant representations are used which capture the group structure of rotations. Their method is trained with the aid of spherical convolutional neural networks (Cohen et al., 2018) based on an auxiliary classification task, this method is unsupervised in the sense that it does not need labels for angles and is capable of estimating relative rotation among different views of an object.

These supervised methods still require the use of labelled data in good quantity and quality. To circumvent these needs, we are interested in the semi-supervised and unsupervised approaches such as Esteves et al. (2018b;a); Rhodin et al. (2018); Kulkarni et al. (2015). Similar to Kulkarni et al. (2015), our focus is to use probabilistic models to capture the topological structure of rotations.

In particular, within the Variational Autoencoders (VAE) Kingma & Welling (2014) context there is a recent growing interest in the study of latent variables with different geometries: *hyperspherical* (Davidson et al., 2018), $SO(3)$ (Falorsi et al., 2018), *hyperbolic* (Mathieu et al., 2019) and *arbitrary closed manifolds* (Pérez Rey et al., 2019; Li et al., 2019a).

VAEs provide an unsupervised method for latent variable estimation and in principle could be used for capturing the underlying geometrical features for a dataset. The Conditional Variational Autoencoder (CVAE) (Kingma et al., 2014) allows to introduce supervision into the latent variable representations.Prokudin et al. (2018) used the CVAE as a supervised method for pose prediction and adding an uncertainty estimation for the input data. This method could not be directly extended to semi-supervised pose estimation without extra modifications.

We use the previous work as a starting point for developing a semi-supervised method for pose estimation based on CVAEs with underlying geometrical latent variables for pose estimation.

## 3 METHODS

### ROTATION REPRESENTATION

It is important to choose an appropriate latent representation for the rotations that captures their topological properties. This has been discussed in Liao et al. (2019) and Mahendran et al. (2018a) where several representations are described. In the case of 2D planar rotations along one fixed axis each rotation can be described in terms of the 1-dimensional circle $S^1$ Liao et al. (2019). Our method will has focus on the use of a probabilistic models (Variational Autoencoders Kingma & Welling (2014) capable of encoding the input images in a circular latent space.

To the best of our knowledge, there are three approaches to implement a variational autoencoder with a hyperspherical latent space. Firstly in Davidson et al. (2018), they have focused on a method that is tailored specifically for hypersphere. On the other hand, Pérez Rey et al. (2019) and Li et al. (2019a) propose methods for choosing arbitrary closed spaces. We have chosen to focus on using the work in Pérez Rey et al. (2019) to represent the rotations in $S^1$ since it can provide arbitrary manifolds as latent space which can be used in further experiments on different applications.

CONDITIONAL VARIATIONAL AUTOENCODERS

The Conditional Variational Autoencoders (CVAE) introduced by Kingma et al. (2014) consist of a method for approximating to the true generative process that produced a dataset in a semi-supervised way. The CVAE assumes a generative process in which the data space $X$ and two latent spaces $R, Z$ are involved. The data space $X$ consists in this case of the set of all possible images with a certain width height and color channels. The latent space $R$ is associated with the variable of interest that we have available in the labelled dataset (in this case the rotation), and the second latent space $Z$ corresponds to an unobserved variable.

In the Variational Autoencoder (VAE) context (Kingma & Welling, 2014), a family of parametric distributions is used to approximate the posterior distributions over the latent spaces. The distribution over the rotations $R$ is denoted by $\mathbb{Q}_{R|x}$ and the distribution over the latent space $Z$ is denoted by $\mathbb{Q}_{Z|x,r}$. Such distributions are denominated encoding distributions and their parameters are calculated with the aid of neural networks. On the other hand a parametric distribution over the data space $X$ is denominated decoding distribution and denoted by $\mathbb{P}_{X|z,r}$.

The generative model is obtained via variational inference by optimizing the parameters of the encoding and decoding distribution in order to maximize the evidence lower bound of the log-likelihood of the available observations (images and rotations). For the labelled dataset the log-likelihood of the image-rotation pairs is maximized. The evidence lower bound for the labelled dataset $\mathcal{L}(x, r) \leq \log P_{X \times R}(x, r)$ for a pair of image and rotation $(x, r) \in X \times R$ is given by

$$\mathcal{L}(x, r) = \mathbb{E}_{z \sim \mathbb{Q}_{Z|x,r}} \left[ \log P_{X|z,r}(x) + \log \left( P_R(r) \right) + \log \left( P_Z(z) \right) - \log \left( Q_{Z|x,r}(z) \right) \right]. \quad (1)$$

Where $P_R$ and $P_Z$ are the chosen prior probability density functions of the prior distributions over the corresponding latent spaces. In the case of the unlabelled dataset the evidence lower bound $\mathcal{U}(x) \leq \log P_X(x)$ takes the form

$$\mathcal{U}(x) = \mathbb{E}_{(z,r) \sim \mathbb{Q}_{Z \times R|x}} \left[ \log P_{X|z,r}(x) + \log \left( P_R(r) \right) + \log \left( P_Z(z) \right) - \log \left( Q_{Z \times R|x}(z, r) \right) \right]. \quad (2)$$

For our experiments, the latent space $Z$ is chosen as a $d$-dimensional Euclidean space $Z = \mathbb{R}^d$ with a normal posterior approximate with diagonal covariance $\mathbb{Q}_{Z|x,r}^{(\mu_Z, \sigma_Z)}$ and a standard normal distribution as prior $\mathbb{P}_Z$ where $\mu_Z$ is the location parameter and $\sigma_Z$ is a vector representing the diagonal of the covariance matrix.

The chosen latent space $R$ that captures the 2D rotations of the object is a circle embedded in $\mathbb{R}^2$ i.e. $R = S^1 \subseteq \mathbb{R}^2$. The encoding distribution $\mathbb{Q}_{R|x}^{(\mu_R, t_R)}$ over $R$ is chosen as in Pérez Rey et al. (2019) where it corresponds to the transition kernel of the diffusion equation over the circle such that $\mu_R$ and $t_R$ correspond to the location and scale parameters respectively. A uniform prior $\mathbb{P}_R$ over $R$ is selected over this latent space. For the decoding distribution $\mathbb{P}_{X|z,r}^{(p_X)}$ we propose a Bernoulli distribution over each of the pixels with parameter $p_X$.

According to Kingma et al. (2014) an extra regression loss term is added to the CVAE objective. This loss term can be considered as the regression loss associated to the posterior over $R$ that estimates the rotations. For an input image $x_l$ and its corresponding true rotation $r_l \in S^1$, the regression loss corresponds to the negative dot product between the estimated location of the posterior approximate $\mu_R(x_l)$ and the true rotation i.e.

$$\mathcal{S}(r_l, x_l) = -r_l \cdot \mu_R(x_l). \quad (3)$$

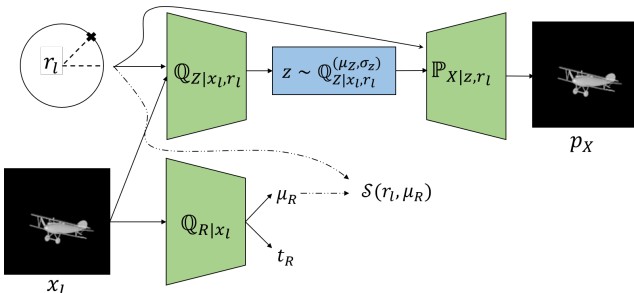

(a) CVAE connections for training using pair of image and rotation. For labelled data, the regression loss takes part in training.

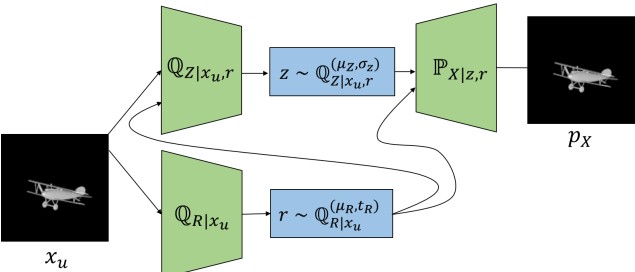

(b) CVAE connections for training using unlabelled images.

Figure 1: Training procedure for CVAE. Green trapezoids represent the encoding and decoding neural networks that calculate the parameters of the corresponding distributions. Blue rectangles represent the sampling of latent variables according to the corresponding reparameterization tricks.

The maximization of the evidence lower bounds for the unlabelled and labelled dataset can be translated into a minimization problem by introducing a minus sign. The minimization loss for the labelled and unlabelled datasets is then described by the following formula

$$\mathcal{J}\left(\{x_l, r_l\}_{l=1}^L, \{x_u\}_{u=1}^U\right) = -\sum_{l=1}^L \left(\mathcal{L}(x_l, r_l) - \mathcal{S}(r_l, x_l)\right) - \sum_{u=1}^U \mathcal{U}(x_u). \tag{4}$$

TRAINING THE CVAE

In order to train a CVAE the labelled and unlabelled data need to be fed to the encoding and decoding distributions respectively. Figure 1 shows how the parameters of the corresponding encoding and decoding distributions are estimated for the labelled and the unlabelled datasets. The sampling from $\mathbb{Q}_{Z|x_l,r_l}^{(\mu_Z,\sigma_Z)}$ is done using the reparametrization trick from Kingma & Welling (2014) while the sampling from $\mathbb{Q}_{R|x_u}^{(\mu_R,t_R)}$ is done by performing a random walk over the circle as described in Pérez Rey et al. (2019).

In order to train jointly the whole dataset of $N = L+U$ unlabelled and labelled images we introduce for each available image $x_i$ an additional supervision value $s_i \in \{0, 1\}$. If the datapoint is labelled then $s_i = 1$ and $s_i = 0$ otherwise. For unlabelled images $x_i$ an arbitrary fixed label $y_i$ is provided. During training the loss function to be optimized corresponds to

$$\mathcal{J}\left(\{x_i, r_i, s_i\}_{i=1}^N\right) = -s_i \sum_{l=1}^N \left(\mathcal{L}(x_i, r_i) - \mathcal{S}(r_i, x_i)\right) - (1 - s_i)\sum_{i=1}^N \mathcal{U}(x_i). \tag{5}$$

This loss function reduces to Equation 4 but provides a practical approach for implementing the training of labelled and unlabelled data.

## 4 DATASET GENERATION AND ARCHITECTURE

In order to test the capabilities of our method we have restricted the use of our method to a controlled setting in which the data annotations and the images provided can be generated arbitrarily. With this purpose we have focused on the use of a dataset of 3D models from which we can generate arbitrary amount of rendered images of objects with known rotations.

ModelNet40 by Wu et al. (2014) is a dataset of 3D CAD models with object instances from forty categories. For the task of pose estimation, an aligned version was provided by Sedaghat & Brox (2015) in which, for each category, all object instances were aligned to a certain frame of reference. From the 3D models we have focused on the airplane class which has the most training instances (626 instances) and used a regular rendering procedures to generate the images for training and testing the CVAE:In the case of the planar 2D rotations of the object, we have generated the views by rendering images from 64 equally distanced angles within the interval $[0, 2\pi]$.

We train with a VGG-like architecture for both the encoder and decoder neural networks similar to the work presented in Prokudin et al. (2018). The encoder networks, for the parameters of $\mathbb{Q}_{R|x}$ and $\mathbb{Q}_{Z|x,r}$, consist of six convolutional layers with 24, 48, 48, 64 and 64 feature channels followed by a fully connected layer of 512 neurons. Each convolution and dense layer has rectifying linear unit activations followed by batch normalization. A final dense layer with linear activations is attached to each network to predict the pertaining parameters to be estimated. To enforce the restriction of positive scale parameters the log-scale parameters are estimated for each encoding density instead.

For the latent space $Z = \mathbb{R}^d$ the dimension is chosen as $d = 50$ with location and log-scale parameters of the same dimension. For the space of rotations $S^1 \subseteq \mathbb{R}^2$ the location parameter is obtained from a 2 neuron dense layer followed by Euclidean normalization. The log-scale parameter is the output of a 1 neuron dense layer.

The decoding neural network consists of the reverse VGG network. First a dense layer with 512 neurons is followed by a combination of up-sampling and convolutional layers with filters 64, 64, 48, 48, 24. The final convolution has a sigmoid activation and returns arrays with same shape as input images.

## 5 EXPERIMENTS

In our experiments we have studied the performance of our approach with respect to two main factors. The amount of labelled data and the coverage of the rotations that are fed to the network.

**Number of renders**    As it was described in the previous section, for each object instance 64 images were generated at different rotations. During training, different subsets of these renders were used to test the effects of the angle coverage. The number of renders used per object instance were $\{64, 32, 16, 8\}$.

**Label percentage**    For testing the effects of changing the amount of available labels we select uniformly randomly a certain percentage of object instances for which we remove all the labels in their corresponding images. We further divide randomly the available labelled images into train(70%) and validation (30%). In each experiment we train for 200 epochs with early stopping and a tolerance of 15 epochs for the loss. The model with the lowest validation loss is chosen as the final model for the given experiment.

### EVALUATION

Evaluation of the rotation regression is done with respect to the rendered images from object instances within the test set of ModelNet40 aligned. For each 3D model instance a set of 64 images at angles equally separated from the interval $[0, 2\pi]$ were generated. All of the trained models are evaluated with respect to the complete test set of 64 images per object instance. For each experiment three repetitions are trained and the relevant metrics are averaged and their standard deviation is reported.

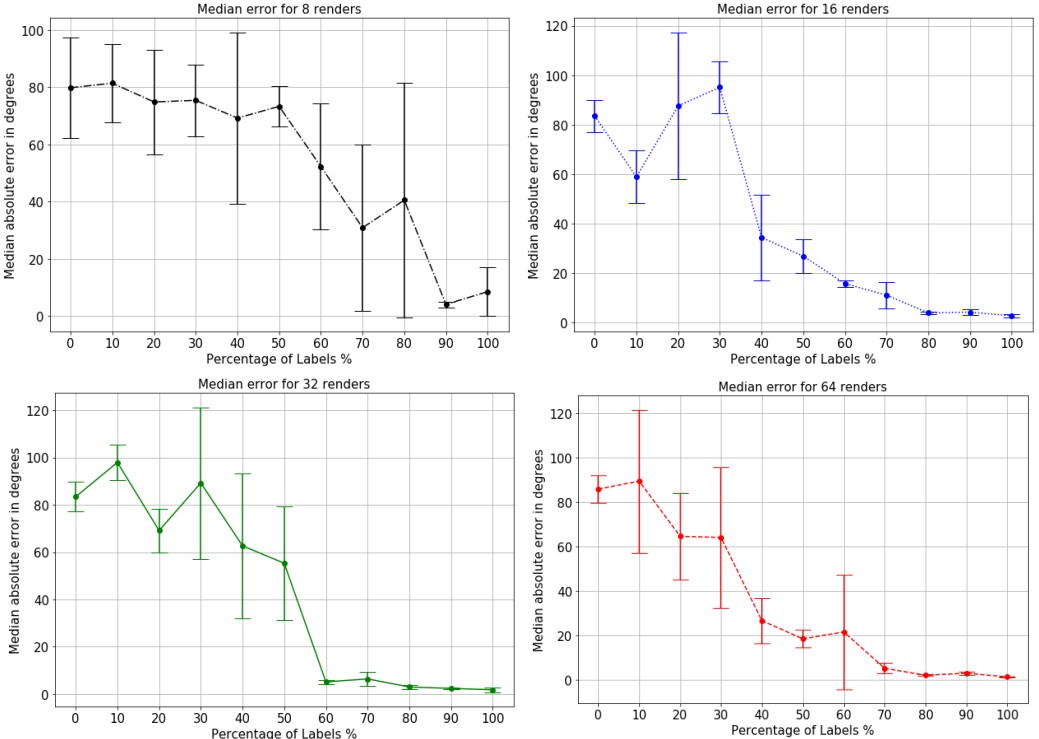

Figure 2: Results for the experiments trained on the regular angle dataset. Median error with respect to the percentage of labelled data, each experiment is averaged over three runs and the corresponding standard deviation bars are shown. Each of the four plots corresponds to the results obtained by training models based on different amounts of rendered images per object instance.

**Median Error**   We consider that the location parameter obtained for an image in the test dataset $\mu_R(x_l) \in R = S^1$ corresponds to the rotation prediction for the labelled image $x_l$ in the test dataset. The angular error between the label rotation $r_l \in S^1$ and the prediction $\mu_R(x_l)$ is calculated as their geodesic distance in $S^1$

$$d_{S^1}(r_l, \mu_R(x_l)) = \arccos(r_l \cdot \mu_R(x_l)). \tag{6}$$

We calculate the median error across the images in the test dataset and report it in degrees.

**Accuracy**   For accuracy, the prediction is considered to be correct if its geodesic distance is smaller than a certain threshold $\theta$, i.e. $d_{S^1}(r_l, \mu_R(x_l)) \leq \theta$. We consider three threshold values $\theta \in \{\frac{\pi}{6}, \frac{\pi}{12}, \frac{\pi}{24}\}$ for our evaluation and report the percentages of correct classifications.

## 6   RESULTS

The results from Figure 2 show the performance of the models with respect to the amount of supervision. A general trend can be distinguished in all plots i.e. increasing the amount of labelled images decreases the median error and thus improves the models' performance. This is an expected result since the models receive more signals that push the training into the correct regression values. It is important to notice that when a relatively small amount of labels is missing (around 80%-90%) the method achieves a similar performance. This means that the method can be used effectively in cases with a moderately restricted labelling of data.

Focusing on the results with respect to the different number of renders per object instance and evaluated with respect to the test images from, it is expected to observe higher variance in the methods trained on less images. For such models, the training has been biased towards a restricted set of views which increases the variability in the predictions for the test set.

It is quite interesting to find that even though the models trained on 8 and 16 renders have seen a restricted amount of views from the object instances, their performance is high when trained on highly labelled data. This means that under high supervision, the amount of example views from an object at different angles can be decreased significantly while mainting competitive performances.

Table 1 shows a comparison of the performance of the method in the fully supervised setting. In such cases the CVAE has a good performance with the median error decreasing with respect to the number of renders provided during training as expected. As it was previously described, the results show that even with few representative samples of an object's rotation (with 8 renders), the method still achieves a low median error and high accuracy (up to $87 \pm 16.3\%$). This performance still comes at the cost of high variance in the resulting models' performance.

| # Renders | Median Error ($°$) | Acc@$\frac{\pi}{6}$(%) | Acc@$\frac{\pi}{12}$ (%) | Acc@$\frac{\pi}{24}$(%) |
|---|---|---|---|---|
| 8 | 8.5±8.5 | 87.0±16.3 | 77.6±27.3 | 65.4±33.7 |
| 16 | 2.6±0.5 | 98.7±0.3 | 96.8±0.8 | 87.6±3.3 |
| 32 | 1.8±1.0 | 99.1±0.2 | 98.3±0.6 | 93.9±4.2 |
| 64 | 1.2±0.3 | 99.4±0.0 | 99.0±0.0 | 97.7±0.1 |

Table 1: Statistics for fully supervised method with 100% of labelled data evaluated on the test dataset. Median error (in degrees) and accuracy percentage at different threshold values is shown.

## 7 CONCLUSION

We have presented a novel approach to the task of semi-supervised pose estimation with limited trained data. The method is capable of receiving a dataset of labelled and unlabelled images. As expected, model's performance increases with the amount of supervision provided to the network. Moreover, when just a small percentage of the training labels are missing the model can achieve comparable performance to that of the fully supervised method. Some future steps to this work include the use of different sampling schemes for generated the rendered views of an object. New sampling schemes would also include arbitrary 3D rotations. Moreover, testing with more challenging datasets such as Pascal3D+ Xiang et al. (2014) would provide a point of comparison with other pose estimation methods..

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
