# OpenReview forum: "Semi-supervised Pose Estimation with Geometric Latent Representations"
_ICLR.cc/2020/Conference — Reject_

### Official Review · AnonReviewer2 · 2019-10-24
**Official Blind Review #2**

**Rating:** 3

**Review:**

This paper tackles the task of rotation estimation in a setting where both labelled and unlabelled examples are available for training. It proposes to learn a generative model of images (a VAE), where the ‘code’ is factored into a latent vector z and the object rotation r.  As in training a VAE, an image encoder that predicts the distribution over (z, r) and the generator are jointly trained, but with additional supervision on the distribution over r for the labelled examples.

I think the overall idea of learning a disentangled generative model in a semi-supervised setting is simple and elegant, and could in principle help leverage unlabelled data. However,  I do have some concerns regarding the specific contributions of this work, and several reservations about the experiments reported, and would overall argue for rejection.

Concerns:
1) The central empirical result stated is that using this approach allows one to reduce amount of labelled data by 10-20 %. First, even if valid, this is not a very convincing reduction in the amount of supervision. However, I feel this claim is not well-established by the experiments:

1a)  The paper should report a baseline with only using the loss in eqn 3 and only training the encoder (using various fractions of training data) to predict the rotation i.e. purely discriminative training without training a generative model. The current plots of performance vs fraction of labelled data don't mean much until compared to a similar plot for this baseline. The current results don't really highlight the importance of training the generative model or using the unlabelled data.

1b) I think there are some inconsistencies in performances reported in Fig 2. I assume the test set is same despite different training data, because the paper states "All of the trained models are evaluated with respect to the complete test set". In this regard, I am puzzled why using 100% labelled data with 16 renders is significantly better than using 50% labelled data with 32 renders -- these should imply similar number of labelled examples, and more unlabelled ones in the former.

2) While the discussion points to this, the paper would really benefit from having results in a real setting, in particular as pose estimation is a field with a lot of prior methods that have been shown to work in these settings. The current results are all in a setup with synthetic, unoccluded data, without background variation, equidistant camera uniformly sampled along a circle. The central idea of using a generative model would be much more difficult to operationalize in a realistic setting where these simplifying assumptions are not made, and I'd only be convinced about the applicability of the approach by results in that setting. As a possible setup, one case use many imagenet images in conjunction with labelled examples in PASCAL3D+ to try this approach.

3) The overall approach maybe novel in context of pose estimation, but this idea of learning a disentangled generative model is not, and there are several papers which do so with varying amount of supervision e.g. see [1] below for similar ideas, and pointers. While some details here may vary, in context of these prior works, I'd view this paper as mostly applying well-established ideas to a new task.

--
In addition to the above, I have a question regarding the training/testing data:
Q: The dataset description only states data was randomly divided - was this random division at an image level, or model level i.e. could different renderings of the same model be in train and test set?
--

[1] Learning Disentangled Representations with Semi-Supervised Deep Generative Models, NIPS 2017. Siddharth et. al.

**Experience Assessment:**

I have published in this field for several years.

**Review Assessment: Checking Correctness Of Derivations And Theory:**

I assessed the sensibility of the derivations and theory.

**Review Assessment: Checking Correctness Of Experiments:**

I carefully checked the experiments.

**Review Assessment: Thoroughness In Paper Reading:**

I read the paper thoroughly.

---

> ### Author Response · Authors · 2019-11-13
> **Reviewer 2 response**
>
> Dear reviewer, I would like to thank you for the time and effort spent in analyzing this paper and for the specific suggestions made to improve this work. We will answer the concerns presented in the review and indicate the future actions to improve the paper.
> 1) The central empirical result stated is that using this approach allows one to reduce amount of labelled data by 10-20 %. First, even if valid, this is not a very convincing reduction in the amount of supervision. However, I feel this claim is not well-established by the experiments:
> 1a) The paper should report a baseline with only using the loss in eqn 3 and only training the encoder (using various fractions of training data) to predict the rotation i.e. purely discriminative training without training a generative model. The current plots of performance vs fraction of labelled data don't mean much until compared to a similar plot for this baseline. The current results don't really highlight the importance of training the generative model or using the unlabeled data.
> A: Thank you very much for your suggestion. We will include these experiments in a future version of the paper.
>
> 1b) I think there are some inconsistencies in performances reported in Fig 2. I assume the test set is same despite different training data, because the paper states "All of the trained models are evaluated with respect to the complete test set". In this regard, I am puzzled why using 100% labelled data with 16 renders is significantly better than using 50% labelled data with 32 renders -- these should imply similar number of labelled examples, and more unlabeled ones in the former.
> A: To answer this question, it is important to clarify that the split between labeled and unlabeled data is done at 3D model level. This means that for 100% labeled and 16 renders the CVAE sees all the available 3D models in the training dataset but only from 16 different angles whereas for the 50% labeled and 32 renders, the model just sees half of the 3D models but with renders that have more angle coverage. Therefore, even though the amount of data is similar in both cases, one case has less examples at model level (which would mean that it has seen less variability in).
>
> 2) While the discussion points to this, the paper would really benefit from having results in a real setting, in particular as pose estimation is a field with a lot of prior methods that have been shown to work in these settings. The current results are all in a setup with synthetic, unoccluded data, without background variation, equidistant camera uniformly sampled along a circle. The central idea of using a generative model would be much more difficult to operationalize in a realistic setting where these simplifying assumptions are not made, and I'd only be convinced about the applicability of the approach by results in that setting. As a possible setup, one case use many imagenet images in conjunction with labelled examples in PASCAL3D+ to try this approach.
> A: Thank you very much for your suggestion, this can further strengthen our claims. We will include these experiments in a future version of the paper.
>
> 3) The overall approach maybe novel in context of pose estimation, but this idea of learning a disentangled generative model is not, and there are several papers which do so with varying amount of supervision e.g. see [1] below for similar ideas, and pointers. While some details here may vary, in context of these prior works, I'd view this paper as mostly applying well-established ideas to a new task.
> A: We agree with the fact that we are using the established framework from [1] in a new setting. One important detail to be taken into account is that we are incorporating prior geometrical knowledge specific to our task in the method. Since rotations have specific topological properties, we have chosen a suitable latent space for the predictions of the network (in this case choosing a circle S^1 as latent space). In a future version of the paper we will try to emphasize this perspective from our work and search for new ways to include more topological properties into our predictions and latent representations.
> 4) In addition to the above, I have a question regarding the training/testing data: Q: The dataset description only states data was randomly divided - was this random division at an image level, or model level i.e. could different renderings of the same model be in train and test set?
> A: The training/testing dataset were divided at model level i.e. during training the CVAE has received no renders from the testing 3D models. This way we ensure there has been no information leakage about certain models from the training dataset into the testing phase.
> Once again, we would like to thank the reviewer for taking the time to analyze the work and provide useful suggestions to improve our work.
> -- [1] Learning Disentangled Representations with Semi-Supervised Deep Generative Models, NIPS 2017. Siddharth et. al.

---

### Official Review · AnonReviewer1 · 2019-10-25
**Official Blind Review #1**

**Rating:** 3

**Review:**

This paper proposes to employ conditional variational autoencoder (CVAE) to estimate the geometry of 2D rotations of objects given images partially labeled. Here, the label represents the geometry of the 2D rotation. The proposed method introduces two latent representation. z is the ordinal latent variable and r a latent representation for the rotations where the latent variable is defined in the 1-dimensional circle in R^2 so that it can naturally represent a hyperspherical latent space.
The construction of the proposed CVAE is straightforward. For labeled images, the (evidence lower bound of the) loglikelihood of the image-rotation pairs is maximized. For labeled images. For labeled images, the (evidence lower bound of the) loglikelihood of the images is maximized.

The decision of the reviewer of this paper is weak reject. The major reason is the lack of technical originality. The construction itself would be novel while each component (e.g., CVAE,  latent representation for the rotations, and semi-supervised construction of CVAE) have been already known.

Experimental results are not surprising but show that the presented method is useful to some extent. In a sense, it is a bit disappointing that we need 50+% images needed to be labeled to achieve < 20-degree error. One interesting observation of this paper is that more labeled images give better results than giving greater number of renders. Expansion to 3D rotations would be a good challenge.



**Experience Assessment:**

I do not know much about this area.

**Review Assessment: Checking Correctness Of Derivations And Theory:**

N/A

**Review Assessment: Checking Correctness Of Experiments:**

I assessed the sensibility of the experiments.

**Review Assessment: Thoroughness In Paper Reading:**

I read the paper at least twice and used my best judgement in assessing the paper.

---

> ### Author Response · Authors · 2019-11-13
> **Reviewer 1 response**
>
> Dear reviewer, I would like to thank you for the time and effort spent in analyzing our work. We would like to provide some answers to the comments made in the review and to make a small clarification with respect to the results.
> 1)	 The construction itself would be novel while each component (e.g., CVAE, latent representation for the rotations, and semi-supervised construction of CVAE) have been already known. Experimental results are not surprising but show that the presented method is useful to some extent. In a sense, it is a bit disappointing that we need 50+% images needed to be labeled to achieve < 20-degree error.
> A: Thank you for your feedback. We will work to provide more novel results in a future iteration of the paper.
> 1)	One interesting observation of this paper is that more labeled images give better results than giving greater number of renders.
> A: We have noticed that our description of the experiments gives rise to certain confusions about the expected results. Take for example the case with 100% labeled and 16 renders vs 50% labeled and 32 renders. The results show that the method with 100% labeled data has considerable better performance even though the number of images is similar which might appear contradictory.
> It is important to clarify that the split between labeled and unlabeled data is done at 3D model level. This means that for 100% labeled and 16 renders the CVAE sees all the available 3D models in the training dataset but only from 16 different angles whereas for the 50% labeled and 32 renders, the model just sees half of the 3D models but with renders that have more angle coverage. Therefore, even though the amount of data is similar in both cases, one case has less examples at model level (which would mean that it has seen less variability in). There is a tradeoff in this case between having more examples from the object instances vs. having more examples of possible angles. We will clarify this point in a future version of the paper.
> 2)	Expansion to 3D rotations would be a good challenge.
> A: Thank you for very much for your suggestion. We will include such expansion in a future version of the paper.
> Once again, we would like to thank the reviewer for the time taken and providing valuable feedback.

---

### Official Review · AnonReviewer4 · 2019-11-05
**Official Blind Review #4**

**Rating:** 1

**Review:**

This paper presents a semi-supervised approach to learn the rotation of objects in an image. The primary motivation is that for rotation estimation datasets may not always be fully labeled, so learning partially from labeled and partially for unlabeled is important. The approach is to use a CVAE with a supervised loss and an unsupervised loss and to jointly train the network. Limited experiments that show performance are presented.

First, the paper solves a very interesting problem with potentially wide applications. The paper is reasonably well-written.

Unfortunately, I don't believe that the contributions of the paper meet the standards of ICLR. I justify my opinion below. The experiments are also very weak.

- While the high level goal of "pose estimation" is clear. Even after reading the paper multiple times, I did not understand the setting well. It appears like the paper looks at the problem of 2D orientation estimation of objects in images. However, this setting is restrictive and not very practical in reality. We mostly care about 3D pose estimation. It would have been good to see results on 3D rotations at the very least.

- Contribution: It is unclear to me what the primary contribution(s) of the paper is. The entire section on CVAE's and losses are quite standard in literature. The interesting part is in combining the supervised and unsupervised parts of the method for the task for pose estimation. But in the end this is a simple weighted loss function (equation 5). So I wonder what is the novelty? What are the new capabilities enabled by this approach?

- Related Work:

Implicit 3D Orientation Learning for 6D Object Detection from RGB Images, ECCV 18


- I would have loved to see a description of the differences in the loss functions (1) and (2). Perhaps this can help elevate the contribution more?

- I also missed justification of why the particular design choice is suitable for this problem? Would direct regression using a simple CNN work better?

- In equation (4), how are the two losses balanced?

- The dataset generation part is just confusing. ModelNet40 is rendered but only 2D rotation is predicted? What does 2D rotation mean for a 3D object?

- Could this method be tested on a dataset like dSprites (https://github.com/deepmind/dsprites-dataset) which has 3D rotations?

- Regarding experiments: I was disappointed to see no comparisons with other approaches or even a simple baseline. A CNN that directly regresses orientation could help put the tables and plots in perspective.

Overall, the problem is important (if lifted to 3D) with important applications. However, the paper does not say anything new about how to solve the problem and the experiments are weak. In its current state, I am unable to recommend acceptance.

**Experience Assessment:**

I have published one or two papers in this area.

**Review Assessment: Checking Correctness Of Derivations And Theory:**

I assessed the sensibility of the derivations and theory.

**Review Assessment: Checking Correctness Of Experiments:**

I assessed the sensibility of the experiments.

**Review Assessment: Thoroughness In Paper Reading:**

I read the paper thoroughly.

---

> ### Author Response · Authors · 2019-11-13
> **Reviewer 4 response**
>
> Dear reviewer, I would like to thank you for the time and effort spent in analyzing this paper and for the specific suggestions made to improve this work. We will answer the concerns presented in the review and indicate the future actions to improve the paper.
> 1)	The entire section on CVAE's and losses are quite standard in literature. The interesting part is in combining the supervised and unsupervised parts of the method for the task for pose estimation. But in the end this is a simple weighted loss function (equation 5). So I wonder what is the novelty? What are the new capabilities enabled by this approach?
> A: With respect to the standard CVAE the choice of the posterior approximate and the prior over the target labels is changed to accommodate the topological properties of the rotations.  In this regard, we have incorporated previous work that explores the use of manifolds as latent spaces.
> As it has been pointed out, one important feature of the method is the possibility to train a regression network that can benefit from both labeled and unlabeled data. Moreover, since the CVAE is a generative model it can be used for view generation of objects although we haven’t focused on this aspect yet.
> Nevertheless, we recognize that we would like to explore more novel ideas that incorporate more prior knowledge of the problem into the pose estimation.
> 2)	I would have loved to see a description of the differences in the loss functions (1) and (2). Perhaps this can help elevate the contribution more?
> A: Both loss functions have been presented in already in the original CVAE paper, the difference lies in the choice of the posterior approximate to the labels and the prior over the label space that incorporates the topological properties of rotations. In a future version of this work we will provide a more detailed explanation on this matter.
> 3)	 I also missed justification of why the particular design choice is suitable for this problem? Would direct regression using a simple CNN work better?
> A: In this problem setting a direct regression would need to take into account first of all the topological properties of rotations. Nevertheless, if such topological properties are incorporated a second problem with using a simple CNN is that it is not completely clear how to deal with unlabeled data. The image reconstruction task within the Variational Autoencoder acts as an auxiliary task that leads the regression network to better rotation predictions when no labels are present.
> 4)	In equation (4), how are the two losses balanced?
> A: We have not yet considered a weighting factor for the loss between labeled and unlabeled data (we present equal contribution from both sides). We are still exploring different approaches for balancing both losses in a principled manner.
> 5)	The dataset generation part is just confusing. ModelNet40 is rendered but only 2D rotation is predicted? What does 2D rotation mean for a 3D object?
> A: We consider a 2D rotation as a rotation with respect to a fixed rotation axis which is shared among same classes of objects. This means that we have restricted the dataset to SO(2) rotations with a fixed rotation axis. In the case of more arbitrary rotations from SO(3) we need to make some extra geometrical considerations for the choice of our latent space of predictions. We will work to extend the work in a future version of the paper.
> 6)	Could this method be tested on a dataset like dSprites (https://github.com/deepmind/dsprites-dataset) which has 3D rotations?
> A: Thank you very much for the suggestion we will incorporate such results in the next version of this paper.
> 7)	Regarding experiments: I was disappointed to see no comparisons with other approaches or even a simple baseline. A CNN that directly regresses orientation could help put the tables and plots in perspective. Overall, the problem is important (if lifted to 3D) with important applications. However, the paper does not say anything new about how to solve the problem and the experiments are weak. In its current state, I am unable to recommend acceptance.
> A: Thank you very much for this suggestion. We will include more experiments in a future version of the paper that try to answer such questions and concerns.
> Overall, we have identified that we are lacking some important experiments to position our method with respect to a baseline. Moreover, our work might lack stronger points of support, we will include experiments with SO(3) rotations and try to explore different directions to improve our contributions. Once again, we would like to thank the reviewer for the time taken into writing a detailed analysis of our work and providing constructive feedback.

---

### Decision · Program_Chairs · 2019-12-19

**Decision:**

Reject

**Comment:**

This paper addresses the problem of rotation estimation in 2D images. The method attempted to reduce the labeling need by learning in a semi-supervised fashion. The approach learns a VAE where the latent code is be factored into the latent vector and the object rotation.

All reviewers agreed that this paper is not ready for acceptance. The reviewers did express promise in the direction of this work. However, there were a few main concerns. First, the focus on 2D instead of 3D orientation. The general consensus was that 3D would be more pertinent use case and that extension of the proposed approach from 2D to 3D is likely non-trivial. The second issue is that minimal technical novelty. The reviewers argue that the proposed solution is a combination of existing techniques to a new problem area.

Since the work does not have sufficient technical novelty to compare against other disentanglement works and is being applied to a less relevant experimental setting, the AC does not recommend acceptance.